# Influence of the Levels of Arsenic, Cadmium, Mercury and Lead on Overall Survival in Lung Cancer

**DOI:** 10.3390/biom11081160

**Published:** 2021-08-05

**Authors:** Sandra Pietrzak, Janusz Wójcik, Piotr Baszuk, Wojciech Marciniak, Małgorzata Wojtyś, Tadeusz Dębniak, Cezary Cybulski, Jacek Gronwald, Jacek Alchimowicz, Bartłomiej Masojć, Piotr Waloszczyk, Darko Gajić, Tomasz Grodzki, Anna Jakubowska, Rodney J. Scott, Jan Lubiński, Marcin R. Lener

**Affiliations:** 1International Hereditary Cancer Center, Department of Genetics and Pathology, Pomeranian Medical University in Szczecin, ul. Unii Lubelskiej 1, 71-252 Szczecin, Poland; baszukpiotr@gmail.com (P.B.); debniak@pum.edu.pl (T.D.); cezarycy@pum.edu.pl (C.C.); jgron@pum.edu.pl (J.G.); aniaj@pum.edu.pl (A.J.); lubinski@pum.edu.pl (J.L.); marcinlener@poczta.onet.pl (M.R.L.); 2Department of Thoracic Surgery and Transplantation, Pomeranian Medical University in Szczecin, ul. A. Sokołowskiego 11, 70-891 Szczecin, Poland; janusz.zenon.wojcik@wp.pl (J.W.); margaretkaw@wp.pl (M.W.); alchim@mp.pl (J.A.); gajic@spwsz.szczecin.pl (D.G.); grodzki@grodzki.szczecin.pl (T.G.); 3Read-Gene, ul. Alabastrowa 8, 72-003 Grzepnica, Poland; wojciech.marciniak@read-gene.com; 4Radiation Oncology Department, West Pomeranian Oncology Center, ul. Strzałkowska 22, 71-730 Szczecin, Poland; bmasojc@gmail.com; 5Independent Laboratory of Pathology, Zdunomed, ul. Energetyków 2, 70-656 Szczecin, Poland; piotr.waloszczyk@zdunomed.pl; 6Priority Research Centre for Cancer Research, Innovation and Translation, Hunter Medical Research Institute, New Lambton Heights, NSW 2305, Australia; rodney.scott@newcastle.edu.au; 7School of Biomedical Sciences and Pharmacy, Faculty of Health and Medicine, University of Newcastle, Callaghan, NSW 2308, Australia; 8Division of Molecular Medicine, Pathology North, John Hunter Hospital, New Lambton, NSW 2305, Australia

**Keywords:** heavy metals, cadmium, lung cancer prognosis, prognostic markers

## Abstract

The effects of heavy metals on cancer risk have been widely studied in recent decades, but there is limited data on the effects of these elements on cancer survival. In this research, we examined whether blood concentrations of the heavy metals arsenic, cadmium, mercury and lead were associated with the overall survival of lung cancer patients. The study group consisted of 336 patients with lung cancer who were prospectively observed. Blood concentrations of heavy metals were measured to study the relationship between their levels and overall survival using Cox proportional hazards analysis. The hazard ratio of death from all causes was 0.99 (*p* = 0.94) for arsenic, 1.37 (*p* = 0.15) for cadmium, 1.55 (*p* = 0.04) for mercury, and 1.18 (*p* = 0.47) for lead in patients from the lowest concentration quartile, compared with those in the highest quartile. Among the patients with stage IA disease, this relationship was statistically significant (HR = 7.36; *p* < 0.01) for cadmium levels in the highest quartile (>1.97–7.77 µg/L) compared to quartile I (0.23–0.57 µg/L, reference). This study revealed that low blood cadmium levels <1.47 µg/L are probably associated with improved overall survival in treated patients with stage IA disease.

## 1. Introduction

Globally 9.6 million people die from cancer every year and lung cancer remains the leading cause of cancer death. Death from lung cancer affects over 1.7 million people annually, which represents 18.4% of the total cancer deaths [1].

Heavy metals such as cadmium, lead, arsenic and mercury play an important carcinogenic role, especially for lung cancer [2,3]

Heavy metals occurring in the environment can be particularly dangerous to human health. The National Priorities List from the Agency for Toxic Substances and Disease Registry (ATSDR), which ranks substances based on a combination of their frequency, toxicity and potential human exposure, includes the four elements we have examined in this study, i.e., arsenic being ranked first on the list, lead second, mercury third and cadmium seventh [4].

Arsenic (As), cadmium (Cd), mercury (Hg) and lead (Pb) are also known as toxic environmental contaminants. The U.S. and international environmental and public health agencies have classified inorganic As and Cd as human carcinogens [5,6]. Sources of exposure to As may be from the air, drinking water and food. This element is a known carcinogen, regardless of whether the exposure occurs through inhalation or ingestion [7]. Arsenic is rapidly absorbed by the digestive tract (70–90%) but has a short half-life (40–60 h), which results in this element being eliminated relatively quickly from the human body [8]. Cd is associated with the development of cancers, other lung and kidney diseases and fetal growth restriction [9,10] but unlike As, it has a long half-life (between 6 and 38 years in the kidney and 4 to 19 years in the liver) as its elimination from the body is extremely slow [11].

Pb is a possible elemental human carcinogen and is a known neurotoxin. Sources of Pb are food and water, gasoline, contaminated dust and actual Pb exposure. According to WHO, there is no level of exposure to Pb that is not harmful [12].

While the main source of Hg is pelagic fish it is also found in water, soils and sediments [13]. Hg and its compounds are very toxic. Elemental Hg is neurotoxic and nephrotoxic, Methyl-Hg compounds are primarily neurotoxic but have also been associated with cognitive impairment in children and fetal abnormalities [14,15].

Despite the fact that the toxicity of heavy metals is widely known, most studies on these elements focus on cancer risk rather than how they may influence survival especially for patients with lung cancer.

The aim of this study was to determine whether concentrations of heavy metals such as As, Cd, Hg and Pb have any influence on the overall survival in lung cancer patients.

## 2. Materials and Methods

### 2.1. Experimental Subjects

A total of 336 patients with a diagnosis of lung cancer were enrolled in the study after providing informed consent. They were recruited at the Department of Thoracic Surgery and Transplantation in Szczecin-Zdunowo Hospital, Poland, between January 2011 and November 2012. This centre treats more than 90% of all patients diagnosed with lung cancer in the West Pomeranian region of Poland. All patients had a confirmed histopathological diagnosis of lung cancer and had undergone surgery. The study was conducted in accordance with the Helsinki Declaration on human research and was approved by the Ethics Committee of the Pomeranian Medical University in Szczecin (Poland) under the number KB-0012/73/10.

### 2.2. Measurement of Heavy Metals Level

#### 2.2.1. Sample Collection and Storage

All patient blood samples were collected at the time of lung cancer diagnosis but before the commencement of any treatment. Consenting patients were asked to fast for at least four hours prior to blood collection. A venous blood sample (10 cm^3^) was collected in tubes certified for the quantification of trace metals (Vacutainer^®^ System, Tube #368381, Becton, Dickinson and Company, Franklin Lakes, NJ, USA). Collected blood was transferred into cryo-vials and stored at −80 °C until analysis. On the day of analysis blood samples were thawed and mixed.

#### 2.2.2. Measurement Methodology

Determination of ^91^AsO^+^, ^114^Cd, ^202^Hg and ^208^Pb was performed using the ICP mass spectrometer ELAN DRC-e (PerkinElmer, Concord, ON, Canada). Before each assay, the instrument was tuned to achieve the manufacturers’ optimum criteria. Oxygen was used as a reaction gas. The spectrometer was calibrated using an external calibration technique. Calibration standards were prepared fresh daily, from 10 µg/mL Multi-Element Calibration Standard 3 (PerkinElmer Pure Plus, Shelton, CT, USA) by diluting with a blank reagent to the final concentration of 0.48; 0.99; 1.98 µg/L for As, Cd and Hg determination and 1; 2; 5; 10 µg/L for Pb determination. Correlation coefficients for calibration curves were always greater than 0.999. Matrix-matched calibration was used. Rhodium was set as the internal standard.

The analysis protocol assumed a 30-fold dilution of serum in a blank reagent. The blank reagent consisted of high purity water (>18 MΩ), TMAH (AlfaAesar, Kandel, Germany), Triton X-100 (PerkinElemer, Shelton, CT, USA), n-butanol (Merck, Darmstadt, Germany), rhodium (PerkinElmer, Shelton, CT, USA), gold (VWR, Steinheim, Germany) and EDTA (Sigma-Aldrich, Leuven, Belgium).

#### 2.2.3. Quality Control

The accuracy and precision of all measurements were tested using certified reference material (CRM), Clincheck Plasmonorm Blood Trace Elements Level 1 (Recipe, Munich, Germany).

Recovery rates were between 80–105% for analyzed elements; calculated recurrency (Cv%) was below 15% for all of the measured elements. The testing laboratory is a member of two independent external quality assessment schemes: LAMP organized by CDC (LAMP: Lead And Multielement Proficiency Program; CDC: Center for Disease Control) and QMEQAS organized by the Institut National de Santé Publique du Québec (QMEQAS: Quebec Multielement External Quality Assessment Scheme).

### 2.3. Statistical Analysis

Study participants were divided into four groups of blood heavy metal levels (quartiles) based on the distribution of heavy metal concentrations in the entire study group. The reference quartile was the one with the lowest number of patient deaths. We observed the study population from the date of diagnosis until death or the 8th of February 2019 with a total follow-up time for our study of 96 months. Data on death were obtained through the Polish National Death Registry. The cause of death from lung cancer was not available, therefore death from any cause was used for the analysis. We modelled the relationship between blood heavy metal levels on overall survival using a Cox proportional hazards analysis. To find potential relationships between levels of heavy metals and overall survival, we performed univariate, as well as multivariate analyses. The multivariate analysis was adjusted for age (continuous), sex, stage (IA, IB, IIA, IIB, IIIA, IIIB, IIIC, IVA, IVB), radiotherapy (yes/no), chemotherapy (yes/no) and smoking status (yes/no). Overall survival in association with heavy metal levels is shown by Kaplan–Meier analysis. All calculations and graphics were performed in an R environment (R: A language and environment for statistical computing; R Foundation for Statistical Computing, Vienna, Austria; R version: 4.0.3).

In order to estimate the cut-off points in relation to the risk of death in the patient group, an analysis of risk curves based on the sliding window method was performed. All samples were sorted by ascending levels of the heavy metal. The ORs were calculated for the deceased/alive and inside/outside sliding window proportion of subjects. The window was sliding one sample per step over the whole range of levels of the elements studied.

In the next step, the calculation sequence is repeated for the sliding window larger by one observation. Calculation window sizes range from 10 observations, up to half of the analysed group size. After completing all calculation sequences, the most frequent concentration is selected (which in each case was considered as the threshold level—according to regression curve fitting).

## 3. Results

The patient characteristics of the study group are shown in Table 1.

The mean level of heavy metals for the entire group was 1.02 µg/L for As (range 0.25–6.69 µg/L), 1.45 µg/L for Cd (range 0.23–7.77 µg/L), 1.02 µg/L for Hg (range 0.01–6.09 µg/L), and 25.29 µg/L for Pb (range 5.91–149.44 µg/L). The mean levels of heavy metals by subgroups including sex, age, smoking status, various treatments, stages and quartiles are shown in Appendix A.

In the multivariate Cox analysis, statistically significant differences were observed for two metals, Cd (quartile III vs. I HR = 1.56; 95% CI 1.02–2.36; *p* = 0.04) and Hg (quartile I vs. IV HR = 1.55; 95% CI 1.03–2.34; *p =* 0.04 and quartile III vs. IV HR = 1.49; 95% CI 0.99–2.22; *p* = 0.05—Table 2). In univariate analysis, the difference among quartiles was significant only for Cd (quartile III vs. I HR = 1.53; 95% CI 1.04–2.27; *p* = 0.03—Appendix A).

We performed a similar analysis separately for each clinical stage of lung cancer. It was only for Cd that a statistically significant relationship between the concentration of this heavy metal in the blood and the length of survival for patients with stage IA lung cancer was observed. This relationship was not observed among the group of other cancer stages (IB-IVB; Appendix A). Mean Cd levels among patients with stage IA lung cancer (*n* = 89) were 1.50 µg/L (range 0.23 to 7.77 µg/L).

Using univariate analysis the hazard ratio (HR) for death from all causes was 2.74 (95% CI 1.15 to 6.50) for patients in the highest quartile of blood cadmium levels compared to those in the lowest quartile. The multivariate model revealed that this relationship had greater significance (HR = 7.36; *p* < 0.01, Table 3). Detailed Cox regression analysis for all factors associated with stage IA lung cancer is presented in Appendix A. The data and results for uni- and multi-variate analysis for the other clinical stages of lung cancer and heavy metals are available upon request.

The detrimental effect of Cd concentration on survival was observed in the group of patients with stage IA lung cancer in the quartile with the highest blood Cd concentrations (Figure 1). The Kaplan–Meier survival estimates according to the quartiles of blood heavy metal levels are presented graphically for all cases in Appendix A.

The cut-off point for Cd concentration for patients with stage IA disease associated with improved survival was 1.47 µg/L (Figure 2).

## 4. Discussion

In a recent study, we suggested that in patients with stage I lung cancer, low serum selenium levels (<69 µg/L) at the time of diagnosis was associated with an increased risk of death (HR−2.73, *p* = 0.01) [16]. In this study we analysed blood concentrations of As, Cd, Hg and Pb in a group of patients with lung cancer. We observed that low Cd levels in patients with stage IA lung cancer are associated with improved overall survival. Patients in the highest [Cd] quartile (blood Cd > 1.97–7.77 µg/L) had a seven-fold increased risk of death during the 96 months of follow-up (HR = 7.36, *p* < 0.01). The cut-off point of Cd concentration was 1.47 µg/L. This suggests that lower blood levels of Cd (below 1.47 µg/L) are associated with improved overall survival in patients with this disease.

Existing data on the effect of heavy metal levels on cancers mainly relate to mortality. In a U.S. study, participants had Cd levels measured in urine samples and corrected for urine creatinine (uCd). Among men, the association between uCd and mortality from lung cancer was statistically significant—19 deaths in Q1-Q3 vs. 112 deaths in Q4 > 0.580 μg/g (aHR = 3.22; 95% 1.26–8.25) [17]. Another cohort study assessed the impact of long-term exposure to Cd (3792 people, 20 years follow-up) and cancer mortality in native North Americans, comparing the 80th and 20th percentiles of Cd. The observed HR was 1.30 (95% CI: 1.09–1.55) for all cancers and 2.27 (95% CI: 1.58–3.27) for lung cancer. Additionally, it was estimated that the percentage of deaths from lung cancer due to smoking that could be attributed to Cd exposure was 9.0% (95% CI: 2.8, 21.8), which translates to a low-to-moderate exposure of Cd being prospectively related to total cancers and lung cancer mortality [18]. However, evidence of the association of Cd with mortality remains unclear. A Japanese study (*n* = 275) did not show a statistically significant association between urinary Cd and cancer mortality [19]. A second prospective Japanese study revealed no association between urinary Cd levels and mortality from lung cancer in men (*n* = 37), although in women (*n* = 19) the hazard ratio for deaths from lung cancer was of borderline significance (HR = 0.64; 95% CI: 0.4–1.1) [20].

Our results are in agreement with a study on 134 patients with nasopharyngeal cancer (NPC), which revealed that a high blood Cd level was a significant prognostic risk factor for NPC progression (HR = 3.76; 95% CI: 1.75–8.06; *p* = 0.001) [21]. In this study, high blood Cd (>3.84 µg/L) was strongly associated with short progression-free survival (PFS) (*p* < 0.001), and short PFS was not associated with other factors such as age at the time of diagnosis, family history of cancer, alcohol consumption, pathological type, type of Epstein–Barr virus antibody (EA, VCA) or Tumor Node and Metastasis (TNM) classification. The results of the current study on the concentration of Cd in the blood of lung cancer patients also showed that a high blood Cd level (>1.47 µ/L) was associated with shorter survival for patients with stage IA disease.

Cd occurs naturally in Zn and Pb ores, it is also part of sewage sludge and a component of phosphorus fertilizers [22,23]. The use of fertilizers in agriculture is one of the main causes of Cd contamination in soil, leading to the accumulation of this heavy metal in plants, animals and humans. Cd has a long half-life (10–35 years), which increases the accumulation of this element in the body and remains in continuous circulation in the environment [22,23,24]. The largest amounts of Cd in food are found in bread (mean ± SD (range): 45 ± 31 µg/week), potatoes and root vegetables (28 ± 11 µg/week). Other vegetables (9.4–8.1 µg/week), as well as seafood, rice, and mushrooms, can also accumulate Cd [25]. Dietary intake of Cd is in the range from 13–35 µg/day or from 0.2 to 0.7 µg/kg body weight for an adult [26]. Industrial workers involved in the production of alloys and batteries, metallurgy, the pigment industry, mining, welders, coppersmiths and demolition workers are exposed to higher amounts of Cd [27,28].

An important source of Cd is tobacco smoking. The content of Cd in tobacco varies, but the typical range is from 1–2 μg/g of dry matter, which corresponds to 0.5–1 μg/cigarette. Moreover, approximately 10% of inhaled Cd oxide is deposited in lung tissues, and from 30 to 40% is absorbed into the systemic blood circulation. Smokers have 4–5 times higher levels of Cd in the blood and 2–3 times higher amounts of Cd in the kidneys than non-smokers [29]. This is particularly hazardous for smokers because Cd has a carcinogenic effect associated with oxidative stress and the weakening of antioxidative defence mechanisms. Cd has been classified as a class 1 carcinogen by the International Agency for Research on Cancer (IARC) despite not being directly mutagenic. Cd inhibits DNA repair and impedes the activity of enzymes involved in removing damage [28,30]. An indirect effect of Cd is the generation of reactive oxygen species (ROS). Excess ROS with reduced antioxidant potential in cells promotes the activation of proto-oncogenes, which culminates in cellular proliferation. Cd substitutes for zinc in the zinc-binding domains of key proteins and ROS invades thiol groups of cysteine residues. One of the pathways involves the activation of c-fos and c-jun transcription factors that together form AP-1, which is responsible for the activation of the proto-oncogene responsible for cell growth and division [31,32,33]. Cd also affects the regulation of cell cycle progression through the activation of certain cellular signals, inhibiting DNA methylation and/or interference with cellular adhesion mediated by cadherin. How Cd affects DNA synthesis and cell proliferation depends on its dose. Inhibition of DNA synthesis and cell division occurs at concentrations above 1 μM. However, a lower level of this element improves DNA synthesis and cellular proliferation [28,34,35]. Moreover, Cd may trigger apoptosis, but can also interfere with the translation process by increasing initiating factors, e.g., TIF3 or elongation factors like TEF-1. Cd can also inhibit DNA methylation, which can result in the excessive synthesis of proto-oncogenes, which are responsible for increased cellular proliferation altering the dynamics of cellular transformation [28,36,37].

Our research did not show statistically significant results for As and Pb levels on overall survival in patients with lung cancer. A U.S. study focused on the relationship between As exposure and bladder cancer mortality from 832 bladder cancer patients, comparing high As exposure (>75 percentile) with low exposure, grouped by As levels as determined from toenail analysis (≤25th) [38]. They observed a prolonged overall survival hazard ratio (HR) of 0.5 (95% CI: 0.4–0.8) (adjustment for age, sex, smoking status, stage, grade and therapy) and high toenail As levels were related to better overall survival in a dose-responsive manner (*p*-trend = 0.004).

A second U.S. surveillance program for male workers with documented blood Pb levels with a median of 12 years of follow-up revealed that the group with the highest blood Pb level (40+ µg/dL) had increased lung and laryngeal cancer standardized mortality ratios—SMRs (1.20, 95% CI: 1.03–1.39, *n* = 174, and 2.11, 95% CI: 1.05–3.77, *n* = 11), respectively. SMR for lung cancer in the highest Pb level group with 20+ years follow-up was 1.35 (95% CI: 0.92–1.90) [39]. While in a South Korean study, female workers with blood Pb levels 10–20 μg/dL and 20 μg/dL had statistically a significant increase in bronchus and lung cancer mortality (RR 10.45, 95% CI: 1.74–62.93 and 12.68, 95% CI: 1.69–147.86, respectively) [40]. None of the studies to date have described the relationship between As or Pb levels and the survival time of patients diagnosed with lung cancer.

In the current study we did not focus on Hg levels since there were inconsistent outcomes under different models of analysis.

Several studies have focused on the effect of Hg on cancer mortality, but none of them investigated the potential correlation between Hg levels and lung cancer. For these reasons, we were unable to conclude whether the results for Hg were an artefact or not. For example Rhee’s et al., 2020 showed that a higher blood Hg concentration is associated with a greater prevalence of non-melanoma skin cancer (NMSC) [41], and not with overall survival. In a study of 6784 male and 265 female workers from four Hg mines and mills in Europe and in male study group, there was no overall excess of cancer mortality [42]. Slovenian and Ukrainian Hg workers did appear to have increased mortality from lung cancer (SMR = 1.64, 95% CI = 1.03–1.38), but exposure to this element was not strongly associated with lung cancer risk and could be the result of co-exposure with radon and crystalline silica [3,42].

Summarizing the current study, we identified a statistically significant result for Cd but not for Pb, Hg or As. The results, overall, revealed that there was a trend in improved survival for patients that had lower Cd levels compared to those that had higher levels of this heavy metal (Appendix A). Further interrogation of this cohort of patients identified the main driver of this trend was a significant difference in the survival of patients with stage IA disease where 90% of patients with the lowest quartile of blood Cd level (<0.57 µg/L) were alive after five years. In comparison, only 43% of patients in the highest quartile (>1.97 µg/L) of blood Cd levels are alive after the five-year follow-up time point. These findings are consistent with experimental data showing that carcinogenic effects of Cd are observed only with higher doses of this element [28,34,35].

Importantly, a large number of people who took part in this project came from the same geographical region and were diagnosed at the same institution. However, there are some limitations of our research.

The most important is a statistically significant result regarding the impact of Cd concentration on survival only for the group of patients with stage IA lung cancer. An explanation for this observation could lie in the differential role of low and high Cd levels in stimulating cell growth and DNA synthesis, which is reflected in the ability of Cd to suppress tumour growth at lower levels and promote it at higher levels [43].

The studied group is not consecutive all lung cancer, but only the less advanced malignancies which have been qualified for surgical treatment. Thus, the number of cases at stage IA is the largest and it cannot be excluded that real data for other stages will be different if more patients are studied.

A further limitation of this study was the data on the causes of death available in our registry, which was not lung cancer-specific. We were unable to determine how many deaths occurred for reasons other than cancer, but the assumption was that most deaths would be due to lung cancer.

Another limitation was the one-time measurement of heavy metals, which might change over time, although all the measurements were unified by the time of blood collection, which was after diagnosis and before treatment.

## 5. Conclusions

In conclusion, the data suggest that low blood Cd concentration (<1.47 µg/L) is associated with improved overall survival in patients with lung cancer. It cannot be excluded that blood Cd levels could potentially be a prognostic biomarker for patients at least with stage IA lung cancer, although this observation needs to be verified on the larger series of patients in different populations.

If data from our studies are confirmed, Cd may become an attractive target for elimination which could improve outcomes for lung cancer patients.

## Figures and Tables

**Figure 1 biomolecules-11-01160-f001:**
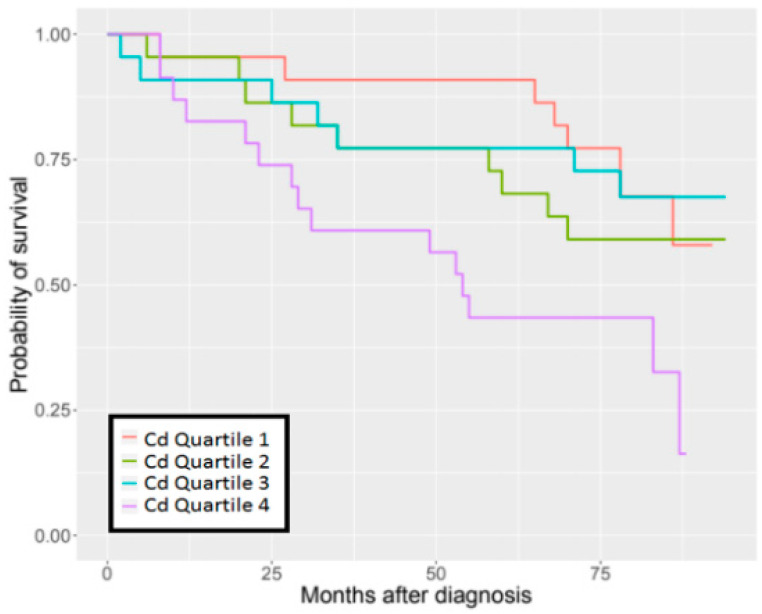
96-months overall survival by Cd levels in stage IA of lung cancer.

**Figure 2 biomolecules-11-01160-f002:**
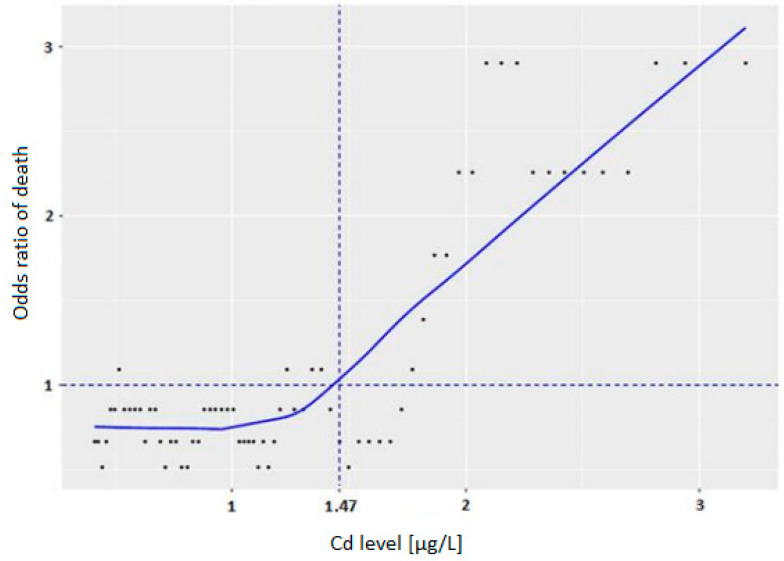
Cut-off point for Cd levels in stage IA disease.

**Table 1 biomolecules-11-01160-t001:** Characteristics of the study group (*n* = 336).

	N	%
**Sex**		
**Male**	222	66.07
**Female**	114	33.93
**Age, mean (range)**	63.71 (43–86)	
**Packyears, mean (range)**	32.59 (0–110)	
**Smoking status**		
**Yes**	315	93.75
**No**	21	6.25
**Stage**		
**I**	153	45.54
**IA**	89	26.49
**IB**	64	19.05
**II**	91	27.08
**IIA**	28	8.33
**IIB**	63	18.75
**III**	76	22.62
**IIIA**	55	16.37
**IIIB**	20	5.95
**IIIC**	1	0.30
**IV**	16	4.76
**IVA**	15	4.46
**IVB**	1	0.30
**Radiotherapy**		
**Yes**	102	30.36
**No**	234	69.94
**Chemotherapy**		
**Yes**	101	30.06
**No**	235	69.94
**Histology**		
**Non-small cell carcinoma**	314	93.45
**Adenocarcinoma**	148	44.05
**Squamous cell carcinoma**	140	41.67
**Large cell carcinoma**	26	7.74
**Combined large cell—small cell carcinoma**	7	2.08
**Small cell carcinoma**	5	1.49
**Other**	9	2.68

**Table 2 biomolecules-11-01160-t002:** Hazard ratios and 95% confidence intervals for various factors on survival from lung cancer depending on As, Cd, Hg and Pb levels.

		Multivariate Cox Regression Models		
Quartile No.	Heavy Metal Level [µg/L]	Hazard Ratio	95% CI	*p*-Value
**As**				
I	0.25–0.60	1.00	-	-
II	>0.60–0.79	1.01	0.68–1.52	0.95
III	>0.79–1.15	1.11	0.76–1.64	0.59
IV	>1.15–6.69	0.99	0.66–1.48	0.94
**Cd**				
I	0.23–0.67	1.00	-	-
II	>0.67–1.13	1.39	0.91–2.11	0.13
III	>1.13–1.86	1.56	1.02–2.36	0.04
IV	>1.86–7.77	1.37	0.89–2.10	0.15
**Hg**				
I	0.01–0.44	1.55	1.03–2.34	0.04
II	>0.44–0.74	1.20	0.80–1.79	0.38
III	>0.74–1.30	1.49	0.99–2.22	0.05
IV	>1.30–6.09	1.00	-	-
**Pb**				
I	5.91–15.57	1.00	-	-
II	>15.57–20.80	1.37	0.90–2.10	0.14
III	>20.80–30.32	1.25	0.81–1.93	0.32
IV	>30.32–149.44	1.18	0.76–1.82	0.47

**Table 3 biomolecules-11-01160-t003:** Hazard ratios and 95% confidence intervals on survival from lung cancer depending on cadmium levels for patients with stage IA lung cancer.

		Univariate Cox Regression Models			Multivariate Cox Regression Models		
Quartile No	Cd Level [µg/L]	Hazard Ratio	95% CI	*p*-Value	Hazard Ratio	95% CI	*p*-Value
I	0.23–0.57	1			1		
II	>0.57–1.11	1.24	0.48–3.21	0.66	3.55	1.02–12.35	0.05 ^1^
III	>1.11–1.97	0.96	0.35–2.66	0.94	2.41	0.64–9.06	0.19 ^1^
IV	>1.97–7.77	**2.74**	1.15–6.50	**0.02**	**7.36**	2.14–25.25	**<0.01**

^1^ Proportional Hazard Requirement is not achieved.

## Data Availability

Data supporting reported results are available from the first author to all interested researchers.

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
