# Peer review of "Influence of the Levels of Arsenic, Cadmium, Mercury and Lead on Overall Survival in Lung Cancer"

_biomolecules, 2021, doi:10.3390/biom11081160_

Round 1

Reviewer 1 Report

In this study the authors investigated the influence of arsenic, cadmium, mercury and lead levels on overall survival in lung cancer. The subject is relevant and the work is on a good level from an experimental and methodic aspect. The authors have also chosen correctly the analysis method (ICP-MS). The manuscript fits within the scope of the journal and will be of interest to many researchers in the field. The data were submitted to statistical analysis to support the statement about the potential relationships between levels of heavy metals studied and overall survival. Moreover, the hazard ratios for various factors on survival from lung cancer depending on As, Cd, Hg and Pb levels was evaluated.

The authors stated that they studied accuracy and precision in section 2.2.3., but they didn’t present the results for these performance parameters according to the certified reference material used.

Author Response

The above parameters have been added to section 2.2.3. (in red)

„Recovery rates were between 80–105% for analyzed elements; calculated recurrency (Cv %) was below 15% for all of the measured elements. The testing laboratory is a member of two independent external quality assessments schemes: LAMP organized by CDC (LAMP: Lead And Multielement Proficiency Program; CDC: Center for Disease Control) and QMEQAS organized by Institut National de Santé Publique du Québec (QMEQAS: Quebec Multielement External Quality Assessment Scheme).”

Reviewer 2 Report

Summary – In this study the authors obtain one blood sampling from patients after lung cancer diagnosis but before treatment. They measure heavy metals including Hg, As, Cd and Pb.  They perform several statistical analyses and identify that the levels of Cd below 1.47 correlate with better survival outcomes, but only in one group (1A stage). While this observation is interesting the conclusions from observation are limited as only one group showed a correlation. No clear explanation for this observation is provided making the results difficult to understand. Further, it is not clear how this study advances the field of lung cancer research and heavy metals. These issues dampen enthusiasm for accepting the manuscript for publication. In addition, the manuscript in its current form has significant writing/grammar issues making the manuscript more difficult to read for this reviewer.

Other issues

Insufficient explanation of why IA stage and high Cd correlate….

Table 3 heading sentence not a complete sentence

Fig 2 Y axis undefined

Incomplete sentence structure, typos and grammar problems throughout the manuscript

A few examples include

Line 51-52 sentence is not complete

140-142 inconsistent order of range in relation to the metal lists

Line 257 typo – two periods

Line 258 incomplete sentence

262 no period – incomplete sentence

304 – need a space between limitation and was

Author Response

Response to Reviewer 2 Comments

Point 1: While this observation is interesting the conclusions from observation are limited as only one group showed a correlation. No clear explanation for this observation is provided making the results difficult to understand. Further, it is not clear how this study advances the field of lung cancer research and heavy metals.

Response 1: We added to discussion comments why results at present are limited to stage IA. Explanations for above observation and potential advances in the field of lung cancer and heavy metal research have been added. (in red)

Point 2: Insufficient explanation of why IA stage and high Cd correlate.

Response 2: Done as mentioned above. (in red)

Point 3: Table 3 heading sentence not a complete sentence.

Response 3: Done. (in red)

Point 4: Fig 2 Y axis undefined.

Response 4: Done. The description of Y axis has been added.

Point 5: The manuscript in its current form has significant writing/grammar issues making the manuscript more difficult to read for this reviewer.  Incomplete sentence structure, typos and grammar problems throughout the manuscript in its current form has significant writing/grammar issues making the manuscript more difficult to read for this reviewer.

A few examples include

Line 51-52 sentence is not complete

140-142 inconsistent order of range in relation to the metal lists

Line 257 typo – two periods

Line 258 incomplete sentence

262 no period – incomplete sentence

304 – need a space between limitation and was

Response 5: The manuscript has been verified. Native speaker Prof. Scott checked the manuscript very carefully. (in red)

Round 2

Reviewer 2 Report

The authors have made substantial changes to the manuscript and it is now acceptable for publication.